# Effect of Silane Coupling Agents on the Rheology, Dynamic and Mechanical Properties of Ethylene Propylene Diene Rubber/Calcium Carbonate Composites

**DOI:** 10.3390/polym14163393

**Published:** 2022-08-19

**Authors:** Weina Bi, Christoph Goegelein, Martin Hoch, Joerg Kirchhoff, Shugao Zhao

**Affiliations:** 1School of Polymer Science and Technology, Qingdao University of Science&Technology, Qingdao 266042, China; 2ARLANXEO Deutschland GmbH Polymer Testing, 51369 Leverkusen, Germany; 3ARLANXEO High Performance Elastomers (Changzhou) Company Limited, Shanghai 200021, China

**Keywords:** silane, interaction, cross-link

## Abstract

The effects of three trimethoxysilanes with different functional groups on the rheology, dynamic and mechanical properties of ethylene propylene diene rubber (EPDM)/calcium carbonate (CaCO_3_) composites were investigated respectively. The results showed that the addition of silane increased the value of M_H_ and M_H_–M_L_ of the compounds. Geniosil XL 33 silane decreased the shear modulus of the EPDM/CaCO_3_ compounds, and the bound rubber content increased slightly with the addition of vinyl trimethoxy silane (VTMS) and methylacryloxy-methyltrimethoxysilane (Geniosil XL 33) silane in the compounds. The vulcanizates with the addition of the VTMS and Geniosil XL 33 silane showed a significant increase in tensile strength and abrasion resistance; however, ethyltrimethoxysilane (ETMS) silane weakened the tensile strength and abrasion resistance of the vulcanizates. At low strain, the cross-linking and reaggregation of fillers resulted in a high storage modulus of vulcanizates with silane. When the strain exceeded 10%, the storage modulus of the vulcanizates with the Geniosil XL 33 and VTMS silane was higher. The loss modulus and tan δ of the vulcanized rubber with the Geniosil XL 33 and VTMS silanes were lower compared to the ETMS and 0 silane.

## 1. Introduction

EPDM is an unsaturated polyolefin rubber with important commercial value. In general, EPDM compounds have excellent chemical resistance to water, ozone, brake fluid, radiation, weather and glycols. EPDM rubber can also be used in cables and wires. Other applications for EPDM include automotive body seals, brackets, roofs, weatherstrips, hoses, conduits, tubing and tires [1].

Mineral fillers, such as calcium carbonate [2], basalt [3], talc [4], mica [5,6] and ontmorillonite [7], are commonly utilized to improve the mechanical properties, dynamic characteristics, wear resistance and processability of rubber composites [8].

Calcium carbonate is an inorganic mineral filler with a wide range of sources; it can be filled in large quantities in rubber to reduce cost, and improve the hardness, rigidity or dimensional stability of the material. CaCO_3_ is treated as a non-petrochemicals based filler with a low CO_2_ footprint. It can also improve the mechanical properties of the material and provide some special properties, such as corrosion resistance, weather resistance, flame retardancy and insulation to the material [9].

The research on EPDM/CaCO_3_ composites mainly focuses on improving impact toughness, such as PP/EPDM/Nano-CaCO_3_ composites [10,11], HDPE/EPDM/CaCO_3_ composites [12] and the reinforcement effect of MAA on nano-CaCO_3_-filled EPDM vulcanizates [2]. There are few studies on the combination of EPDM/CaCO_3_ exclusively [13,14]. The main reason is because calcium carbonate did not have a reinforcing effect in EPDM; the different surface or interfacial properties between calcium carbonate and the matrix [15] always resulted in a low bond strength between the filler and the rubber [16]. In order to improve the interaction between the filler and the matrix, titanate, aluminate, a silane coupling agent and phosphate are mainly used for the surface treatment of calcium carbonate [17,18].

For CaCO_3_, the interaction with silanes is less known and it is assumed it will not react with silanes due to the absence of surface-hydroxyl groups.

Zoltan Demjen [19] et al. found that γ-methacryloxypropyltrimethoxysilane (MPTMS) molecules were parallel to the CaCO_3_ filler surface; the long alkyl chains of saturated carboxylic acid derivatives and stearic acid adsorbed tightly on the CaCO_3_ filler surface in the form of a single molecular layer by dissolution experiments. MPTMS improved its adhesion on the CaCO_3_ surface and formed a surface layer that was not easily soluble. According to Ishida and Koenig [20], there was a small amount of Ca(OH)_2_ and other oxide and hydroxide impurities on the CaCO_3_ surface; this reacted with the organic functional groups of the silanes. Ishida and Miller [21] suggested that 20% of the MPTMS adsorbed on the CaCO_3_ surface was due to the reactive surface impurities.

Nagata, Takahashi et al. [22] studied the reactivity of alkoxy groups in silane coupling agents on inorganic surfaces and analyzed the structure of the silane covering layers on the particle surfaces. The study showed that there was no interaction between the CaCO_3_ particle surface and γ-mercaptopropyltrimethoxy silane; the silane-treated CaCO_3_ filler particles did not contribute to the enhancement of the vulcanized rubber.

Most of the current research is focused on nano-calcium carbonate [23,24]. The CaCO_3_ nanoparticles were treated as follows: first, CaCO_3_ was dispersed in anhydrous ethanol; after the silane coupling agent was hydrolyzed, CaCO_3_ nanoparticles with a certain silane coupling agent were added together to react; after the reaction, the products were filtrated and washed with anhydrous ethanol to remove the unreacted modifiers; and finally, the silane modified CaCO_3_ nanoparticles were obtained [24]. The process is complex.

In this paper, micron calcium carbonate was used as the only filler, and silane was simply added to calcium carbonate without any treatment before mixing. Thus, the modification process is simple; this is very beneficial for the composites processing.

This study attempts to provide a theoretical basis for the effect of silanes in EPDM/CaCO_3_ composites. The rheology properties, bound rubber content, mechanical properties and dynamical mechanical properties were conducted to determine the effect of silane on the EPDM/CaCO_3_ compounds and vulcanizates.

## 2. Materials and Methods

Ethylene-propylene-diene monomer rubber (EPDM) was supplied by ARLANXEO with the trade name Keltan 6950C. Calcium carbonate (CaCO_3_) particles have an average particle size of 1.7 μm; the specific surface area is 4 m^2^/g, which was obtained from Omya GmbH with the trade name Omyacarb 1-AV. Another CaCO_3_ (stearic coated) with the trade name Omyacarb 1T-AV was also from Omya GmbH. Vulkanox ZMB2/C-5 and Vulkanox HS/LG from Lanxess were used as anti-aging agents. The vulcanizing agent BIPB was purchased from AkzoNobel Polymer Chemistry. Extender oil (PARALUX 6001R) was obtained from Chevron. The silane coupling agent vinyl trimethoxy silane (VTMS) was supplied by Evonik Industries AG; methylacryloxymethyltrimethoxysilane (Geniosil XL 33) was from Wacker Chemie AG; ethyltrimethoxysilane (ETMS) was from ABCR GmbH. The formulation shown in Table 1 was used in this study.

The EPDM underwent a mastication process by using an internal mixer for 30 s. Carbonate (silane was mixed into the filler before mixing) was added into the internal mixer and it was further mixed for 3 min. The rotor speed was adjusted to increase the mixing temperature, and the mixing was continued for 5 min after the power curve reached a stable level; finally, the rubber was discharged at 130 °C. The vulcanizing agent and other compounding agents were added on the open mill (roll temperature 50 °C, rotating speed 20 r/min); the compounds were mixed homogeneously and then discharged. After 24 h, compression molding was carried out using a hydraulic steam-heated press from Werner&Pfleiderer GmbH. The test specimens were vulcanized in a steel mold; 2 mm and 6.3 mm thick rubber slabs were compression molded respectively for times equivalent to t_90_ + 10% and t_90_ + 25% minutes plus a mold specific pre-heating time, at 120 bar clamping pressure and a temperature of 175 °C. The slabs were pressed between PTFE foils.

Vulcanization characteristics were determined using a Moving Die Rhometer MDR2000 (Alpha Technologies Company, Akron, OH, USA) at 175 °C, according to ASTM D 2084-95.

A tensile test was carried out according to the ISO-37 standard with dumb-bell S2 test piece using a universal testing machine from Zwick/Roell 2010 Co., Germany, at a speed of 500 mm/min. The length and width of the ends of the samples used were about 75 mm and 12.5 mm, respectively. The gauge length of the specimen used was 25 mm.

In order to obtain the tear sample for the tear test, the rubber compound was compressed using a square mold of 2 mm in thickness. The samples were then left at room temperature for at least 6 h before they were cut into trouser shape. The tear sample was tested using a universal testing machine from Zwick/Roell 2010 at room temperature with the crosshead speed of 50 mm/min in accordance with DIN 53507.

Resistance to wear is one of the most important properties of a rubber compound. Wear is usually considered in terms of abrasion, which is defined as the loss of material that results from mechanical action on the rubber surface. According to DIN 53516, cylindrical test specimens (diameter 16 mm, thickness 6 mm) are measured by the Zwick 6101 (Germany) abrasion testing machine. The abrasive resistance is calculated from the following formulae:(1)Abrasive resistace=100×volume loss of sample

Dynamic rheological measurements of unvulcanized rubber compounds were carried out using a rubber processing analyzer RPA 2000 (Alpha Technologies, Akron, OH, USA), with a strain range of 0.28–100% in shear mode at 60 °C and a frequency of 1 Hz.

Simple shear specimens are exposed to force-controlled cyclic loading using a servo hydraulic MTS 831 elastomer test system (MTS Co., MI 55344-2290, USA) with a strain dynamic amplitude range of 0.2–82.9 mm/mm at 1 Hz and temperature of 60 °C. The sample geometry corresponds to a cylindrical sample with a diameter of 20 mm and thickness of 6 mm.

### Bound Rubber Analysis

To evaluate the bound rubber (BdR) of the rubber composites, 0.5 g of uncured rubber was cut into pieces and located in a steel cage having a mesh of 400. The cage was immersed in toluene for 2 days where the solvent was renewed each day. The samples were dried up in a vacuum oven at 100 °C; the bound rubber was calculated based on the following formula, where m_0_ is the initial rubber mass and m_dis_ is the amount of dissolved rubber in grams:(2)$$\mathrm{BdR}=\frac{m_{0}-m_{\mathrm{dis}}}{m_{0}}\times 100\%$$

## 3. Results and Discussion

### 3.1. Rheology Properties of EPDM/CaCO_3_ Compounds

From Table 2, it can be seen that the addition of silane has little effect on the minimum torque of the sample, while the maximum torque of the sample increases slightly after the addition of silane. The effect of silane on the M_H_–M_L_ value of the samples is obvious. The order of the M_H_–M_L_ value of the samples is as follows: Geniosil XL 33 > VTMS > ETMS > 0 silane. The addition of silane reduces TS1 and the optimum curing time, especially for the Geniosil XL 33 silane.

The torque variation was in accordance with the cross-linking density of the samples. The maximum torque reflects the cross-linking density of the rubber. The M_H_–M_L_ value was used to assess the cross-linking density of the EPDM rubbers qualitatively [25], and is related to the tensile strength and elongation of the rubber. As can be seen from Figure 1 and Table 2, the results showed that the maximum torque of the specimen with silane added is significantly higher than that without silane; moreover, the M_H_–M_L_ values increased with the addition of silane. When the torque exceeded 10 Nm, the effect of silane on the torque was manifested, especially in Geniosil XL 33 and VTMS. It was concluded that when the vulcanization level is close to 50% (M_10Nm_/M_H_), more cross-linkage appears in the rubber added with the VTMS and Geniosil XL 33 silane.

### 3.2. Effect of Silane on CaCO_3_ Dispersion in EPDM/CaCO_3_ Compounds

As can be seen from Figure 2, the shear modulus of all the compounds decreased gradually with the strain amplitude increasing. Compared to others, the shear modulus of the compounds with the Geniosil XL 33 (line in green) silane is relatively low in the low strain region (0.1–10%).

The decrease in the shear modulus of EPDM/CaCO_3_ compounds with the addition of the Geniosil XL 33 silane may occur due to two reasons: first, the acryloyloxy groups in the Geniosil XL 33 silane are polar, and its polarity is related to the strength of silane adsorption on the calcium carbonate surface [19]; and second, the carbon chain in the molecule of Geniosil XL 33 is longer in comparison to VTMS and ETMS. When it adsorbed on the surface of calcium carbonate, it provided enough space between calcium carbonate particles; thus, this increased the distance between the calcium carbonate particles and decreased the force between the calcium carbonate particles.

Therefore, the addition of the Geniosil XL 33 silane slightly improved the dispersibility of calcium carbonate in the EPDM rubber.

### 3.3. Bound Rubber Analysis of EPDM/CaCO_3_ Compounds

Table 3 shows the bound rubber content of EPDM/CaCO_3_ compounds with different silanes. It can be seen that the amount of bound rubber for the compound without silane is about 3%. The amount of bound rubber was slightly increased by the addition of the VTMS and Geniosil XL 33 silane. It is considered that the addition of the VTMS and Geniosil XL 33 silane increased the amount of the molecular chains of rubber with restricted mobility that were adsorbed on the surface of calcium carbonate; it also improved the interactions between the filler and the rubber matrix.

Scholars who supported the Payne effect dominated by filler-rubber interactions believed that the mechanism of the Payne effect is related to the adsorption and desorption of molecular chains with restricted mobility on the filler surface under stress-induced effects, which is closely related to the formation of bound rubber. From Figure 2, it was known that the Payne effect is slightly reduced in compounds with the addition of the Geniosil XL 33 silane. Therefore, it is considered that the bound rubber in EPDM/CaCO_3_ compounds originated from the adsorption of rubber molecular chains on the surface of calcium carbonate.

### 3.4. Mechanical Properties of Vulcanized Rubber

It can be seen that the tensile strength of the EPDM/CaCO_3_ vulcanizates was significantly improved with the addition of Geniosil XL 33 and VTMS from Figure 3. The contribution of the Geniosil XL 33 silane to the tensile strength of the vulcanizates was more obvious. The higher the tensile strength of the vulcanizates, the lower the elongation at break. The addition of the ETMS silane weakened the tensile strength of the vulcanized rubber compared to the vulcanizates without silane.

Table 4 below showed the tear resistance and abrasion properties of the vulcanizates. The addition of the VTMS and Geniosil XL 33 silane has little effect on the tear properties. But the tear strength and abrasion resistance showed a decrease when ETMS was added. When VTMS or Geniosil XL 33 was added, the abrasion property of the vulcanizates was obviously improved, especially the Geniosil XL 33 silane.

The interaction or entanglement between alkyl chains and macromolecules in ETMS could be easily overcome during stretching or tearing. The lack of chemical bonding between ETMS and the rubber macromolecule chains was the main reason for the low mechanical strength of the vulcanized rubber [26]. So, the decoupling of the filler from the rubber network was observed in Figure 2 when ETMS was used. The Geniosil XL 33 is a special silane; maybe, it can even activate the CaCO_3_. We can see its benefit in DIN abrasion in Table 4.

Table 5 showed a comparison between the mechanical properties of different CaCO_3_ filled EPDM/CaCO_3_ composites. The formulation in Table 5 is the same as the specimen with 0 silane in Table 4. It was found that the tensile strength, tear resistance and abrasion resistance in Omyacarb 1-AV (without stearic coated) filled EPDM composites are better than in the Omyacarb 1T-AV filled ones.

### 3.5. Dynamic Mechanical Properties of EPDM/CaCO_3_ Vulcanizates

As shown in Figure 4a, the storage modulus G’ of the EPDM/CaCO_3_ vulcanizates gradually decreased with the increase of dynamic strain amplitude in the range of 0.02~83% strain. All the vulcanizates exhibited a significant Payne effect. The storage modulus of the vulcanizates with silane was higher than that without silane when the strain was less than 10%. The storage modulus was increased due to the cross-links between the silane and macromolecules in the rubber matrix. The thermal history during the vulcanization assembled the fillers and decreased the filler dispersion in the rubber matrix [27]; this also increased the modulus of the vulcanizates.

When the strain exceeded 10%, the storage modulus of the vulcanizates with the Geniosil XL 33 and VTMS silane were much higher. The filler network and filler-polymer interactions were gradually disrupted at high strain, while the cross-linked network and hydrodynamic effects were at work. Therefore, the possible reasons for the high storage modulus are: silane was involved in the cross-linking reaction with the rubber macromolecule chains, which increased the cross-link density of the vulcanized rubber; and silane played a role connecting the filler and rubber macromolecules, which increased the filler-rubber interaction [28]. These results are in a good agreement with part 3.1, which discussed the rheology properties.

The relationship between the loss modulus G″ and the dynamic strain amplitude is shown in Figure 4b; when the strain is lower than 10%, the loss modulus–strain curve of each vulcanizate varied irregularly. However, when the strain is over 10%, it can be seen that the loss modulus of the vulcanizates decreased slightly with the strain increasing; furthermore, the loss modulus of the vulcanizates with silane addition is lower than that without silane addition. The loss modulus of the vulcanized rubber with the Geniosil XL 33 silane is the smallest, followed by the VTMS silane. The loss modulus of the vulcanized rubber slightly decreased when the strain exceeded 20%, which was due to the weakness of the instantaneous recovery and reconstruction ability of the filler network at large strain. Loss modulus is the energy loss during the process of breaking and reconstruction of the filler network, which leads to a lower G″ of the vulcanized rubber at large strain [28].

According to the views of Kraus [29] and Wang [30] on the filler–filler interaction, the loss modulus is closely related to the breaking and reconstruction of the filler network structure. Thus, the decrease of the loss modulus is attributed to the increase of filler dispersion; i.e., the weakening of the filler network.

It was known from Figure 2 that the addition of the Geniosil XL 33 silane slightly reduced the aggregation of fillers; it weakened the filler network structure of the final vulcanized rubber compared to that without silane, and lowered the loss modulus of the vulcanized rubber with the Geniosil XL 33 and VTMS silanes.

The loss factor (tan δ) of the vulcanizates with a dynamic strain amplitude curve is shown in Figure 4c. In the strain range of 0.28–10%, the tan δ of each vulcanizate varied irregularly. When the strain exceeded 10%, the tan δ of the vulcanizates without and with the ETMS silane slightly increased with the strain increasing; however, the tan δ of the vulcanizates with the VTMS and Geniosil XL 33 silane decreased at high strain. This is because the addition of the VTMS and Geniosil XL 33 silane enhanced the cross-linking network of the vulcanizates compared to those without silane and with the ETMS silane. The rubber molecular chains were more prone to breakage and slip when the strain was high; this resulted in increased friction and heat generation, which lead to a higher loss factor than those with the VTMS and Geniosil XL 33 silane [31].

## 4. Conclusions

(1)The M_H_ and M_H_–M_L_ values increased with the addition of silane. When the vulcanization level is close to 50%, more cross-linkage appears in the rubber added with the VTMS and Geniosil XL 33 silane.(2)Geniosil XL 33 improved the dispersibility of CaCO_3_ in the EPDM matrix.(3)The addition of the VTMS and Geniosil XL 33 silane improved the interaction between the filler and the rubber matrix. The cross-links between the VTMS or Geniosil XL 33 silane and macromolecules in the rubber matrix contributed to the increase in the tensile and abrasion properties of the vulcanizates, which have a good agreement with the M_H_ and M_H_–M_L_ values in respect of rheology properties. When using the ETMS silane, although EPDM is non-polar, the filler–rubber interaction becomes minimal. In addition, the mechanical properties of the composites were obviously decreased compared to no silane.(4)The storage modulus of the EPDM/CaCO_3_ vulcanizates gradually decreased with the increase of dynamic strain. At low strain, a high storage modulus of the vulcanizates was obtained with silane addition. When the strain exceeded 10%, the storage modulus of the vulcanizates with the Geniosil XL 33 and VTMS silanes was higher; this resulted from the greater cross-linkage that occurred between the silane and rubber macromolecule chains. Compared to the ETMS and 0 silane, a lower loss modulus of the vulcanized rubber with the Geniosil XL 33 and VTMS silanes was obtained due to the weakening of the filler network; while tan δ of the vulcanizates with the VTMS and Geniosil XL 33 silanes also decreased.

In typical EPDM, compounding carbon black and CaCO_3_ are often used together. The main function of CaCO_3_ is to lower the volume costs. The optimal compositions for such composites depend on the intended application; many different filler systems can be used.

CaCO_3_ gives very little contribution to strength and reinforcement. Mineral fillers need surface activation, mainly with silanes, for bonding into non-polar rubber matrices. Indeed, silane has a benefit when used for CaCO_3_ in EPDM compounds. In this study, CaCO_3_ was combined with EPDM (Keltan ECO grade); thus, the work may also contribute to green compounding.

## Figures and Tables

**Figure 1 polymers-14-03393-f001:**
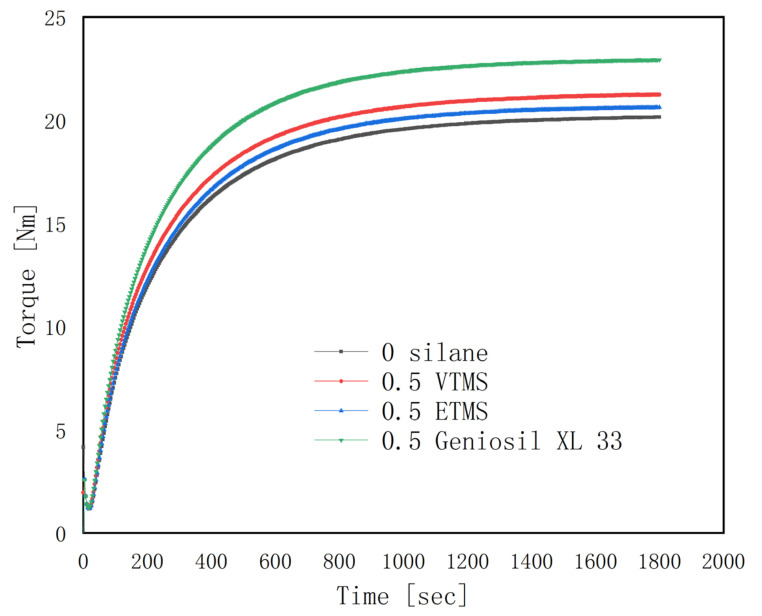
Vulcanization curve of the EPDM/CaCO_3_ composites.

**Figure 2 polymers-14-03393-f002:**
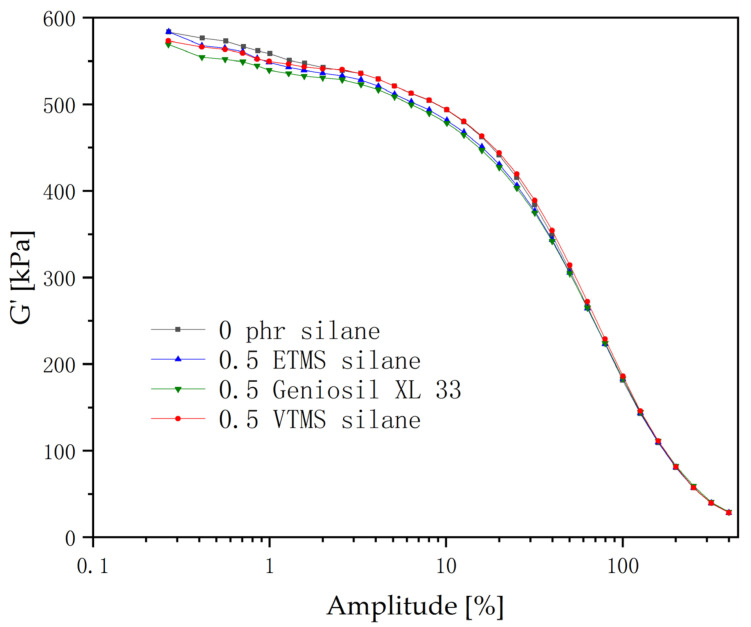
Relationship between the shear modulus and strain of EPDM/CaCO_3_ compounds.

**Figure 3 polymers-14-03393-f003:**
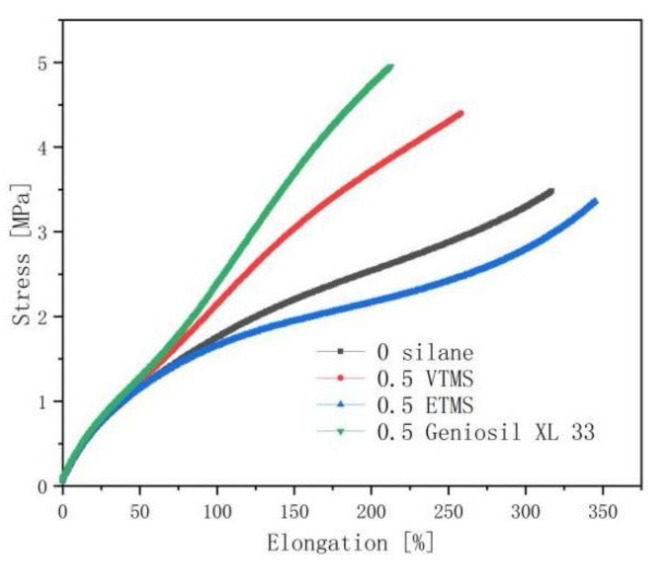
Stress–strain curves of the EPDM/CaCO_3_ composites.

**Figure 4 polymers-14-03393-f004:**
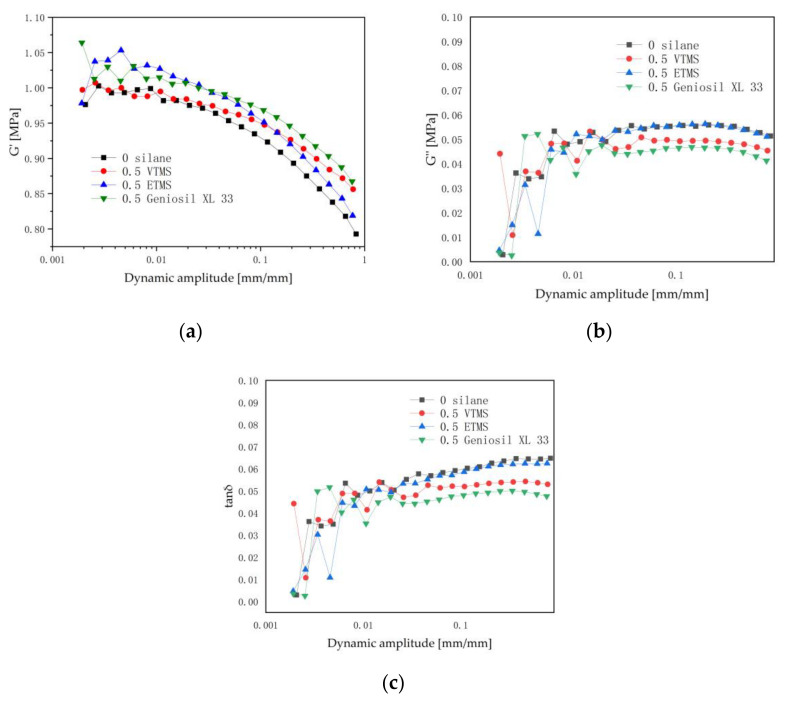
The curves of G′, G″ and tan δ versus strain for the EPDM/CaCO_3_ vulcanizates: (**a**) storage modulus (G′); (**b**) loss modulus (G″); and (**c**) loss factor (tan δ).

**Table 1 polymers-14-03393-t001:** Recipe used to prepare EPDM/CaCO_3_ compounds; the values are in Phr (parts per hundred rubber).

Ingredients	Dosage
EPDM	100
CaCO_3_	100
BIPB	7
ZMB2/C-5	1
TMQ	0.75
PARALUX 6001R	25
Silane	0/0.5

**Table 2 polymers-14-03393-t002:** Rheometer parameters of the EPDM/CaCO_3_ composites.

Silane	0	VTMS	ETMS	Geniosil XL 33
M_L_/Nm	1.17	1.18	1.17	1.22
M_H_/Nm	20.15	21.26	20.63	22.95
M_H_–M_L_/Nm	18.98	20.08	19.46	21.73
T_s1_/sec	37.2	34.2	36	33
T_90_/sec	619.92	608.52	610.62	590.82

M_H_: the maximum torque, M_L_: the minimum torque.

**Table 3 polymers-14-03393-t003:** Bound rubber content of the EPDM/CaCO_3_ blends.

Silane	0	VTMS	ETMS	Geniosil XL 33
BdR, %	3.12	3.46	3.16	3.36

**Table 4 polymers-14-03393-t004:** Mechanical properties of the EPDM/CaCO_3_ composites.

Silane	0	VTMS	ETMS	Geniosil XL 33
Tear resistance, N/mm	1.4	1.5	1.2	1.5
Abrasion volume, mm^3^	593	508	745	377

**Table 5 polymers-14-03393-t005:** Comparison of mechanical properties in the different CaCO_3_ filled EPDM/CaCO_3_ composites.

Filler/ Trade Name	CaCO_3_ (Uncoated)/Omyacarb 1-AV	CaCO_3_ (Stearic Coated) Omyacarb 1T-AV
Tensile strength, MPa	3.3	2.7
Elongation at break, %	300	235
Tear resistance, N/mm	1.4	1.2
Abrasion volume, mm^3^	593	775

## Data Availability

All the data are available within this manuscript.

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
