# Peer review of "Effect of Silane Coupling Agents on the Rheology, Dynamic and Mechanical Properties of Ethylene Propylene Diene Rubber/Calcium Carbonate Composites"

_polymers, 2022, doi:10.3390/polym14163393_

Round 1

Reviewer 1 Report

This paper presents an analysis of the properties of composites (EPDM) / (CaCO3) under the conditions of using different coupling agents

 The paper may be of interest to the scientific community.

The authors can consider the following aspects:

- The introduction needs to be substantially improved considering other current bibliographic sources (only 3 bibliographic sources are from the last 5 years). Also, at the end of the introduction, the research objectives and the structure of the paper must be presented more clearly;

- A better characterization of the materials and technologies used to make the test pieces is required. For example, you need to specify the parameters of the vulcanization process and the equipment used. It is also necessary to specify the dimensions of the samples made and macroscopic images for them. At the same time, it is necessary to explain the decision to consider only the composite variant containing 100 phr CaCO3.

- It is necessary to detail the research methodology and, in particular, the section on the choice of analysis methods used in research;

- The results obtained are not relevant in the conditions in which a microscopic analysis of the samples is not presented, which would allow a correlation between the microscopic structure of the composite and its properties;

- The discussion part must be much developed in order to highlight the novelty brought by the research presented in the paper in relation to other research in the field. In this form, it is not possible to identify the novelty of research compared to previous research;

- Conclusions should be more concrete and future research directions presented.

Author Response

Dear editor:
I would like to thank the editor for giving us a chance to resubmit the paper, and also thank the reviews for giving us constructive suggestions which would help us both in English and in depth to improve the quality of the paper, 
Here I submit a new version of “Effect of Silane Coupling Agents on Dynamic and Mechanical properties of Ethylene Propylene Diene Rubber/Calcium carbonate composites” ,which has been modified according to the reviews’ suggestions. Efforts were also made to correct the mistakes and improve the English of the manuscript.
   I make all the changes in red in the revised manuscript. Please find the attached file for details.
Sincerely yours,
Weina Bi

Reviewer 2 Report

The manuscript "Effect of Silane Coupling Agents on Dynamic and Mechanical properties of Ethylene Propylene Diene Rubber/Calcium carbonate composites" is well written. there some points needs clarifications.

1. By adding silane compounds the authors says that more cross-linkage appears in rubber. Did the authors verify in what extent such happen.

2. From results in Tables as well figures there no standard deviation shown. Did the authors test only one sample or several? Is the reproducibility of the results given

3. If using such composite of rubber, which applications can be drawn from it and are there optimal compositions. Please verify such.

4. It would be informative adding a Table on end to compare those effect in strength to other rubber fillers applied before. Please include those

5. Did the authors made SEM and how does rubber composite changed, are they more rough in morphology? Which application would such composite aimed for. Please include that in conclusion

Author Response

(The authors gave the same response as above.)

Round 2

Reviewer 1 Report

The authors revised their manuscript according to my suggestions. Thus the manuscript can be accepted for publication.